# Polish Multi-Institutional Study of Children with Ependymoma—Clinical Practice Outcomes in the Light of Prospective Trials

**DOI:** 10.3390/diagnostics11122360

**Published:** 2021-12-14

**Authors:** Aleksandra Napieralska, Agnieszka Mizia-Malarz, Weronika Stolpa, Ewa Pawłowska, Małgorzata A. Krawczyk, Katarzyna Konat-Bąska, Aneta Kaczorowska, Arkadiusz Brąszewski, Maciej Harat

**Affiliations:** 1Department of Radiotherapy, Maria Sklodowska-Curie National Research Institute of Oncology Gliwice Branch, 44-101 Gliwice, Poland; 2Department of Pediatrics, Medical University of Silesia, 40-752 Katowice, Poland; amizia-malarz@sum.edu.pl (A.M.-M.); wstolpa@gczd.katowice.pl (W.S.); 3Department of Oncology and Radiotherapy, Faculty of Medicine, Medical University of Gdansk, 80-210 Gdansk, Poland; ewa.pawlowska@gumed.edu.pl; 4Department of Pediatrics, Hematology and Oncology, Medical University of Gdansk, 80-210 Gdansk, Poland; mkrawczyk@gumed.edu.pl; 5Wroclaw Comprehensive Cancer Center, 53-413 Wrocław, Poland; konat.katarzyna@dco.com.pl; 6Department of Oncology, Wroclaw Medical University, 53-413 Wrocław, Poland; 7Department of Children Oncology and Haematology, Wroclaw Medical University, 53-413 Wrocław, Poland; kaczorowskaa@usk.wroc.pl; 8Department of Neurooncology and Radiosurgery, Franciszek Lukaszczyk Memorial Oncology Center, 85-796 Bydgoszcz, Poland; braszewskia@co.bydgoszcz.pl (A.B.); haratm@co.bydgoszcz.pl (M.H.); 9Department of Oncology and Brachytherapy, Nicolas Copernicus University, Collegium Medicum, 85-067 Bydgoszcz, Poland

**Keywords:** ependymoma, children, brain tumours, spinal tumours, radiotherapy, chemotherapy, pediatric oncology

## Abstract

We performed a multi-institutional analysis of 74 children with ependymoma to evaluate to what extent the clinical outcome of prospective trials could be reproduced in routine practice. The evaluation of factors that correlated with outcome was performed with a log rank test and a Cox proportional-hazard model. Survival was estimated with the Kaplan–Meier method. The majority of patients had brain tumours (89%). All had surgery as primary treatment, with adjuvant radiotherapy (RTH) and chemotherapy (CTH) applied in 78% and 57%, respectively. Median follow-up was 80 months and 18 patients died. Five- and 10-year overall survival (OS) was 83% and 73%. Progression was observed in 32 patients, with local recurrence in 28 cases. The presence of metastases was a negative prognostic factor for OS. Five- and 10-year progression-free survival (PFS) was 55% and 40%, respectively. The best outcome in patients with non-disseminated brain tumours was observed when surgery was followed by RTH (+/−CTH afterwards; *p* = 0.0001). Children under 3 years old who received RTH in primary therapy had better PFS (*p* = 0.010). The best outcome of children with ependymoma is observed in patients who received radical surgery followed by RTH, and irradiation should not be omitted in younger patients. The role of CTH remains debatable.

## 1. Introduction

Ependymomas are central nervous system tumours that can grow from the ependymal cells within the ventricles of the brain or spinal canal. The majority of them (around 70%) arise in the infratentorial compartment in the posterior fossa, with 25% diagnosed in the supratentorial parts of the brain and the remaining in the spinal canal [1,2]. Since the first diagnosis of ependymoma in 1924 by Bailey P., much has been discovered, but important treatment issues still require clarification [1,2,3,4]. The identification of those who can benefit from adjuvant treatment and the sequence of such therapy are still the main questions [5]. Although the Children’s Oncology Group trial ACNS0121 confirmed that gross total resection or near-total resection, followed by immediate adjuvant radiotherapy (RTH), can provide satisfactory results in children with brain tumours, both over and under 3 years old [6,7,8]. The benefit of adding chemotherapy (CTH) into ependymoma treatment and the prognostic ability of histopathological grading criteria to risk-stratify patients are still both inconclusive and contentious. The prognostic role of molecular profiling and treatment implications are still a matter of clinical trials [9]. As prospective studies regarding the treatment of children with ependymoma are available, the authors believe that a comparison of their outcome with a multi-institutional clinical outcome could provide valuable information regarding how these results are reproduced in clinical practice [6,7,10,11,12,13,14,15,16,17,18,19,20]. Thus, the aim of our study is to assess the long-term results of the treatment of children with ependymoma in a multi-institutional cohort of paediatric patients and to identify factors related to better outcome regarding the role of CTH, RTH and the best sequence of adjuvant therapy.

## 2. Materials and Methods

A retrospective analysis of children treated due to ependymoma in 6 institutes (one hospital in Bydgoszcz, one in Gdańsk, one in Gliwice, one in Katowice and two in Wrocław) in Poland between 1971 and 2019 was performed. The cooperation between RTH departments and children’s oncologic hospitals was established in order to conduct this research. All consecutive patients younger than 18 years old who were diagnosed with ependymoma during the study period and had at least 2 months of follow-up from the date of diagnosis were included into the study. In all cases, the diagnosis was based on the pathologic examination of the tumour tissue samples collected during a biopsy or surgery and diagnostic imaging (computed tomography, CT and, in later years of the study, magnetic resonance, MR). The medical systems in each participating centre were screened individually to identify patients meeting the study criteria, collected and then all the collected records were re-checked and verified by the first author. The study was approved by the ethical committee in MSC National Research Institute of Oncology in Gliwice (number: KB/430-05/20) and performed according to the Helsinki Declaration.

Study endpoints included: overall survival (OS), which was calculated from the date of diagnosis to the date of death or last follow-up, and progression-free survival (PFS), calculated from the date of diagnosis to the date of the progression of the tumour or death of the patient. Progression was evaluated with diagnostic imaging unless the patient presented with clinical signs of progression of the disease. The Eastern Cooperative Oncology Group (ECOG) scale was used to classify patients’ performance status at the time of diagnosis.

Parameters included into analysis were age, tumour histopathological grade, the date of primary diagnosis, the given curative treatment, patients’ performance status, the extent of resection (EOR), the total dose of RTH, irradiated volume (craniospinal/local), disease symptoms, tumour location (supra/infratentorial and cranial/spinal), time between the surgery and irradiation and the date of progression and death. The Polish National Cancer Registry was contacted to obtain the missing dates of deaths. All those various patients’ characteristics were included in the univariate and multivariate analysis to identify their impact on OS and PFS. Patient outcomes were divided into 4 categories: alive with no evidence of disease, alive with disease, dead of concurrent/intercurrent disease and dead of unknown cause.

In the statistical analysis, the Kaplan–Meier method was used to estimate OS and PFS. Median follow-up was calculated by the Kaplan–Meier analysis with the reversed meaning of the status indicator. Comparisons were made with the use of the log-rank test. The Cox proportional-hazards model for the univariate and multivariate analyses of the prognostic factors were applied. Variables with *p*-value of <0.05 in the univariate Cox analysis were used in the multivariate Cox analysis. Statistical analysis was performed using Statistica software (version 13.3.721.1, Stasoft Polska TIBICO Software Inc. Palo Alto, CA, USA).

## 3. Results

### 3.1. Group Characteristics

The study group consisted of 74 consecutive patients and the characteristics of them are presented in Table 1. Tumours were diagnosed in 17 children under the age of 3 years old. Patients with spine tumours were usually older than those with brain lesions (median age of 13 vs. 7 years old). The majority of patients were diagnosed based on MR, except those treated in the early years of the study. Dissemination at diagnosis was present in seven cases. The majority of patients (83%) showed good performance status. The mean and median tumour dimensions were: 43 × 36 × 34 mm and 40 × 35 × 34 mm, respectively. The most common symptoms were nausea or vomiting, present in 36 cases (49%), and headaches (34 patients, 46%). The most common symptom in cases of patients with spinal tumours was paresis, observed in five out of eight cases. In nine patients, information about disease signs was not available. Myxopapillary ependymomas were solely diagnosed within the spinal canal, while ependymoma G II was the most common subtype in this location. Anaplastic tumours were the most diagnosed brain lesions in our group, both in supra- and infratentorial space. The majority of patients had histopathologic diagnosis of ependymoma confirmed by a second independent neuropathologist.

### 3.2. Primary Treatment

All patients had surgery at the beginning of primary treatment. MR confirmed radical (R0) resection in 20 patients (27%). The second and third surgery was performed in 19 (26%) and 5 cases (7%), respectively. During the reoperation, a more aggressive subtype of ependymoma was discovered in eight cases (grade III). Surgery was the only treatment applied in 13 cases (18%). In 32 patients, the surgery was followed by CTH and then by RTH, concurrent with or followed by, in some cases, systemic treatment. In three patients, surgery and adjuvant CTH were the only treatment methods applied. The surgery was followed by irradiation in 26 cases, and, among them, six received systemic therapy afterwards. In total, 32 children received CTH prior to RTH and 9 after RTH. Looking solely at those with spine lesions, R0 surgery was performed in five cases, R1 in two and, in one patient, information regarding the surgical outcome was unavailable. In Poland, the majority of children’s oncologic centres apply CTH according to the national protocols. The standard treatment in patients with cranial grade III ependymoma included: maximal safe resection followed by four cycles of CTH, adjuvant RTH and eight more cycles of CTH. Although due to the long study period, varied treatment protocols were employed, and some patients were treated differently. However, almost 74% received systemic therapy with etoposide 150 mg/m^2^ and ifosfamide 3 g/kg (delivered every three weeks for 4 to 12 cycles). In patients with cranial grade II tumours and spinal tumours without dissemination, standard treatment included surgery, followed by RTH.

RTH as part of the primary treatment was applied in 58 cases, and, in all but one, was delivered with radical intention. One six-year-old patient received local palliative irradiation with fraction dose (fd) of 1.25 Gy to total dose (td) of 30 Gy in the very early years of the study. The characteristics of the radical RTH delivered during primary treatment are presented in Table 2. Those who had craniospinal irradiation received td of 30.0 to 36.8 Gy (median 36 Gy) with median fd of 1.5 Gy on craniospinal field and boost on the tumour/tumour bed with margin to td of 43.2–59.31 Gy (median 54 Gy). Local RTH was delivered with a median fd of 1.8 Gy to td dose within the range of 45 to 60 Gy (median 54 Gy). One patient had radiosurgery (SRS) with a single fraction of 24 Gy on the tumour field with margin. Among those with spinal tumours, RTH was applied in six cases with a median fd of 1.67 Gy to a median td of 45 Gy.

In all but five cases, which were irradiated before the year 2000, 3D imaging was used in the treatment planning. An individual thermoplastic mask was used for each patient for treatment position reproducibility. A CT scan was done in the treatment position for treatment planning. All patients who had craniospinal irradiation were treated in prone position with two or three masks: one covering the head and one or two (the number depended on patient’s height) covering the region of the chest and pelvis. The treated volumes and normal structures were defined on CT with MR fusion for the patients treated since the introduction of the Eclipse planning system in year 2003 (or later, depending on the hospital in which children were irradiated). Fusion with preoperative and postoperative CT and MR (if available) was used in the treatment planning process to help visualize the tumour and the tumour bed, also since the year 2003. The gross tumour volume (GTV) included the tumour bed and the residual lesion, as defined by the contrast-enhanced T1-weighted, T2-weighted and fluid-attenuated inversion recovery (FLAIR) sequences. In the pre-MR-era, a preoperative and postoperative contrast-enhanced tumour visible on CT was considered GTV. Local and craniospinal irradiation were performed in 29 and 26 cases, respectively. CTV in cases of patients who received local RTH covered: tumour bed with margin in 21 cases or tumour bed with remaining lesion in 6 cases. In two patients, due to the extensive involvement of ventricles, the irradiation of the ventricular system and tumour bed with margin was performed. An additional geometric expansion was added to the above-mentioned CTV to create the planning target volume (PTV) in case of set-up errors and intra- and infrafraction motion.

Among 17 children under the age of 3 years old, seven did not receive RTH as part of primary treatment. In the remaining ten, the time from surgery to the beginning of RTH ranged from 30 to 391 days, with a mean and median of 143 and 116 days, respectively. The mean and median time from surgery to the beginning of RTH in children older than 3 years old was 115 and 110 days, respectively.

The primary treatment of eight patients with spinal tumours consisted of surgery alone in two, surgery combined with RTH in three of them and surgery followed by CTH and then RTH in three cases. Among them, two patients with grade II tumours presented with metastases at diagnosis.

### 3.3. Recurrence of the Disease

The progression of the disease was observed in 32 patients (2 with spinal tumours, 30 with cranial lesions), and local recurrence was the most common site of failure, observed in 28 cases. In four patients, local and distant relapse was diagnosed, and, in five, distant metastases were found during the follow-up. The most common site of the metastases was the brain, in eight patients (four of them had relapse in the ventricular system and two multiple metastases also in the spinal canal) and spinal canal in two patients. All but one received treatment, with surgery performed in 25 cases (78%), RTH delivered in 24 and CTH in 17 patients. Two patients with local recurrence of spinal tumours had surgery that was followed by RTH in one case. Both are alive, one with disease and one with no disease signs 12 years after the recurrence treatment.

Among those who received irradiation, 8 patients had craniospinal RTH and 15 had local therapy, and, in one patient, RTH treatment field description was unavailable. Conventionally fractionated RTH and SRS/stereotactic RTH was delivered in 18 and 5 patients, respectively. Td and fd used in conventional irradiation ranged from 30 to 61.05 Gy (median 54 Gy, mean 51.8 Gy) and 1.5 Gy to 2 Gy (median 1.8 Gy, mean 1.71 Gy), respectively. In seven patients, it was the second course of RTH. SRS was delivered with a median fd of 10 Gy and td ranged from 5 to 24 Gy.

### 3.4. Survival Analysis

Median follow-up was 80 months. During that time, 18 patients died (14 died with disease, and, in 4, the cause of the death was unknown); 13 are alive with disease, and 43 are alive with no signs of the disease. One, 2-, 5- and 10-year OS from the date of diagnosis was 98%, 95%, 83% and 73%, respectively. Figure 1 presents the OS and PFS of the whole group (Figure 1).

Factors that were found to have statistically significant negative impact on OS in univariate analysis are the presence of metastases at diagnosis (*p* = 0.003), grade III subtype (*p* = 0.042) and worse performance status (*p* = 0.023). Patients after R0 resection tended to have better OS, although the difference was not statistically significant in the univariate analysis (*p* = 0.068). Multivariate analysis confirmed that the presence of metastases was the only factor independently affecting OS (*p* = 0.0001). Other analysed factors (age, tumour location, primary treatment applied or the sequence of adjuvant treatment, factors related to RTH, disease symptoms) were not found to influence OS.

Survival analysis of a subgroup of patients without metastases showed that patients with grade II tumours tended to live longer (*p* = 0.053, 10-year OS of 89% vs. 69% for grade II vs. grade III). Although the difference was not statistically significant, patients with spinal tumours had much better 10-year survival of 87%, compared to 73% for patients with brain lesions. The site of metastases (brain +/− spine vs. only spine) also did not reach statistical significance; however, those who had only spinal metastases had 5- and 10-year OS of 100% compared to 62% and 31% for the brain metastases group, respectively.

Median PFS was 77 months. One, 2-, 5- and 10-year PFS was 95%, 75%, 55% and 40%, respectively. Univariate analysis showed that the use of RTH in the primary treatment had positive impact on PFS (*p* = 0.001), with a trend for better survival in patients with brain tumours without CTH preceding RTH (*p* = 0.058). The trend for better outcome after R0 resection was also observed in terms of PFS (*p* = 0.078). Other factors (age, performance status, tumour grade, tumour location, primary treatment applied or the sequence of adjuvant treatment, factors related to RTH, symptoms of the disease) were not found to be statistically significant in terms of PFS.

An additional analysis based on the grade groups was performed. Univariate and multivariate analysis confirmed that, among patients with grade III tumours, RTH and R0 resection were found to improve PFS (*p* = 0.001 and *p* = 0.005), with no impact of the dose escalation above 54 Gy and no difference between craniospinal or local irradiation. Also, adding CTH was not found to affect PFS (*p* = 0.39). Looking at the sequence of treatment, the worst results were observed in surgery or surgery + CTH only groups, with the best outcome in patients with surgery followed by RTH and then adjuvant CTH (*p* = 0.014, Figure 2).

Progression-free survival analysis of a subgroup of patients with brain tumours without metastases confirmed the positive impact of RTH on the outcome, with 10-year PFS of 50% vs. 10% for RTH and no-RTH group (*p* = 0.001). There was no difference in the outcome between the low- (below 54 Gy) and high-dose groups. Similarly, we did not find a difference with regard to the extent of the field (local vs. craniospinal). Looking at the sequence of treatment, the best results were observed in patients with surgery followed by RTH (+/− CTH afterwards) (Figure 3, *p* = 0.0001).

An additional analysis of children under the age of 3 years old was performed. The observed 5- and 10-year OS were 86% and 71%, respectively. A much worse outcome was observed in terms of PFS, with 5- and 10-year PFS of 54% and 30%. RTH was part of the treatment in 10 of 17 patients, and those who received irradiation as part of the primary therapy had much better PFS (*p* = 0.010, Figure 4). RTH was the only treatment-related factor that had an impact on survival in this subgroup of patients.

The 5- and 10-year OS of patients with spinal tumours was 87%. Tumour progression was observed in three cases, with local progression observed in two and distant in one patient. The 5- and 10-year PFS was 70%. Recurrence was treated with surgery in one, surgery and RTH in one, and one patient did not receive any form of treatment. All patients who had recurrence of the disease had grade II tumours after non-radical surgery, combined in two cases with adjuvant RTH. The median PFS of the non-RTH group was 22 months, compared to 69 months for the RTH group.

Looking at the treatment of recurrence, patients who received RTH as part of the treatment of the relapse had better 5-year OS of 85% vs. 65% for the non-RTH group, but the difference was not statistically significant (*p* = 0.38). There was no difference observed in terms of OS depending on the treatment field used (craniospinal vs. local).

Acute and late toxicity evaluation was not performed in all of the patients, so the results were not included in this analysis. Despite that, the information concerning secondary tumours was collected, and, in four patients, secondary meningioma within the irradiation field was diagnosed. Among them, two patients are alive, and two died due to an unknown cause. All of those patients were irradiated in the years 1972–1993 with old RTH techniques, which could be one of the reasons for such an RTH side-effect. Td delivered ranged from 30 to 54.8 Gy, with two patients irradiated on craniospinal fields and one on the local field, and, in one patient treated in the year 1985, the information concerning RTH was unavailable.

## 4. Discussion

The management of children with ependymoma is still a controversial topic in paediatric neuro-oncology and may vary among institutions, from only-surgery to a combination of surgery, RTH, CTH or even autologous hematopoietic stem cell transplant (autoHSCT, Table A1) [10,11,12,13,14,15,16,17,18,19,20,21,22,23,24,25,26,27,28,29,30,31,32,33,34,35,36,37,38,39,40,41,42,43,44,45,46,47,48,49,50,51,52,53,54,55,56,57,58,59,60]. Over the last few years, the diagnostics of brain tumours has been reconsidered and the emphasis has been put on molecular profiling and the search for markers that would help the selection of patients for particular treatment [33,61,62,63,64,65,66,67]. This reflects the last 2021 edition of the WHO Classification of Tumors of the Central Nervous System [61]. According to the newest nomenclature, ependymomas are divided into supratentorial, infratentorial and spinal, based on the localization of the primary lesion, and then are subdivided by genetic features into three different, partly age-related, and prognostically relevant groups each: spinal subependymoma, spinal myxopapillary ependymoma, spinal ependymoma, posterior fossa subependymoma, posterior fossa ependymoma-A, posterior fossa ependymoma-B, supratentorial subependymoma, supratentorial ependymoma YAP1-fused and supratentorial ependymoma RELA-fused [33]. This genetic data, combined with clinical and histological features, was used for developing a risk-stratification model for posterior fossa ependymoma that could potentially serve as a model for future treatment decisions [65]. Studies regarding the predictive role of the molecular features of spinal tumours have recently been published and could provide a foundation for future therapeutic strategies [66]. However, so far, the bleak picture of therapy contrasts with the burgeoning knowledge of ependymoma biology [68]. Prospective clinical trials reported the outcome with various treatment protocols applied; however, clinical practice outcomes do not necessary reproduce the results reported in those trials, and our analysis aimed to investigate the differences and similarities with regard to the treatment outcome. Additional analysis of possible prognostic and predictive factors was also performed.

### 4.1. Surgery

Surgery plays a critical role in the treatment of patients with ependymoma, and it is clear that the extent of resection impacts the outcome, and the best results are observed in patients with R0 resection irrespective of tumour location, cranial or spinal [6,7,8,9,10,11,12,13,14,15,16,17,18,19,20,22,23,24,26,27,33,34,37,39,40,54,58]. The goal of radical surgery could be achieved during the reoperation without compromising the benefit of R0 resection, which was confirmed in the ACNS0121 trial; however, this result concerns only patients with brain tumours [6]. Patients with brain tumors arising from the floor and the lateral aspect of the fourth ventricle were found to have worse outcome than those arising from the roof, possibly due to the more difficult surgery, if the rate of postoperative deficits is kept as low as possible [67]. The persistent lack of agreement is probably a result of the fact that the vast majority of failures occur inside the tumour bed or in the region treated with a high dose, especially when radical resection was not achieved [33,44,68]. The important role of radical surgery is also relevant to the treatment of recurrence and was found to improve the outcome by several authors [33,68,69].

### 4.2. Radiotherapy

Postoperative RTH has been widely accepted as a necessary part of adjuvant treatment, but uncertainties remain regarding the optimal dose, fractionation, volume and timing [43,44,45]. The first study of Merchant et al. changed the paradigm of target volume from craniospinal to local irradiation and demonstrated the value of the td being increased to 59.4 Gy at the primary tumour site [36]. After that, various attempts have been made to improve the outcome. Massimino et al. in their prospective study investigated whether the use of hypofractionated RTH instead of conventionally fractionated could impact survival with comparable outcome in both arms of the study despite high, 70.4 Gy in 1.1 Gy fractions, dose delivered twice daily in a hypofractionated regimen [11]. Hypofractionated schema (60 Gy delivered after complete resection and 66 Gy after partial resection in two daily fractions of 1 Gy) used in the study of Conter C. et al. also did not show the benefit of this RTH fractionation over conventional therapy [12]. In the second AIEOP protocol, Massimino et al. investigated the role of an additional boost of 8 Gy in two fractions delivered in case of the presence of a residual tumour after RTH with a td of 59.4 Gy. The results of patients who received such irradiation—5-year PFS and OS of 58% and 69%—were encouraging, since in all of those cases, complete removal of the tumour was not possible [10]. Moreover, other authors confirmed that increasing the td in conventional fractionation up to 59.4 Gy or more and better coverage of the target volume with the aim of increasing the dose delivered to the highly radioresistant regions could be an effective way of reducing local failures [23,44,46]. Our results are in accordance with the studies that confirmed the benefit of immediate adjuvant RTH. However, we did not find a benefit of dose escalation, probably due to several reasons: the number of patients who received such treatment was quite low (11 cases), often combined with craniospinal irradiation, and the median td was only 55.8 Gy, much lower than the current recommendations. Immediate, adjuvant local RTH to the tumour bed with margin with a td of 59.4 Gy, considering the tolerance of nearby structures, in children over 2 years old is strongly recommended. A td of 54 Gy is advocated in younger patients. The fusion of pre- and post-surgical MR with planning CT and the use of irradiation techniques like intensity-modulated radiation therapy (IMRT) or volumetric modulated arc therapy (VMAT) could lower the possibility for future neurocognitive deficits due to more precise treatment planning and delivery. Craniospinal fields should be reserved for cases with dissemination of the disease.

Moreover, the use of proton therapy is of particular value because of very conformal dose delivery, no exit dose and, as a consequence, the potential for increasing the td without additional toxicity. The study published so far showed that this method of irradiation is safe and effective, but the impact of higher td was proven only in terms of local control [37,53,54]. The outcome of patients treated with proton RTH in Sato et al.’s study also confirmed this hypothesis, with 3-year PFS of 82%. The diagnosis of radiation necrosis requiring medications was observed in 8% of patients (six children—in three cases after proton therapy and in three after IMRT), and two more presented vascular-related side effects [39]. Further studies, which in a prospective way will evaluate the benefits and risks of proton therapy on large group of patients, are warranted.

The role of RTH as part of the treatment of recurrence has been investigated by many authors [68,70,71,72,73,74,75,76,77,78,79]. The use of stereotactic RTH or radiosurgery with a total dose of 9 to 36 Gy in one to few fractions, depending on the location of the recurrent tumour, the age of the patient, the time from the first RTH and the previous doses delivered to the nearby organs at risk, with encouraging results [70,72,74,75,76,77,78,79]. The comparison of the reirradiation results with non-irradiated patients showed the benefit in survival [77]. However, the best results were observed if repeated RTH followed repeated surgery [70,74,75,76,77,78,79]. Conventional RTH with the use of craniospinal irradiation in cases of distance recurrence was described by Merchant et al., but achieving good local control came at a cost of side effects that were more pronounced than in the stereotactic arms [76]. Moreover, the French society of children’s cancer reported a series of patients reirradiated with local conformal technique to a median td of 57 Gy using conventional or hypofractionated regiments. Despite very high doses delivered in the second RTH, no toxicity of grade 2 or higher has been reported, except for one case of radiation necrosis after radiosurgery [79]. In the biggest series of 101 reirradiated patients, the 10-year cumulative incidence of grade 3 or higher radiation necrosis was 8%, although more than half of the group received craniospinal irradiation as part of the recurrence treatment [75]. So far, there is no consensus on the reirradiation modalities, neither on dose, nor fractionation, nor technique. However, repeated RTH of recurrent paediatric intracranial ependymoma is established as a safe, effective treatment that may affect long-term survival in selected patients. Decision should be made individually, and repeated surgery before reirradiation could further improve the outcome [70,74,75,76,77,78,79].

### 4.3. Systemic Treatment

The role of CTH was evaluated in several prospective and retrospective studies. The introduction of new therapeutic agents or a new combination of already used ones, or dose intensification with combination of autoHSCT, was the intent of the majority of them, with one idea—to avoid or delay the irradiation. RTH was left as the treatment of progression and eventually was used in the majority of patients in all of those studies [13,14,15,16,17,18,19]. As showed in Table A1, the outcome of children treated in those studies in all but one was inferior compared to the trials or analyses that did not defer the irradiation (Table A1). However, the one, the study of Grundy et al., did not report the evaluation of the response to systemic therapy nor the assessment of neurocognitive abilities, which could change as a result of prolonged CTH [15,47,48]. For this reason, most North American institutions have abandoned deferral strategies that use CTH, and adjuvant RTH is considered the best standard of care for all children from the age of 12 months onwards [47]. Garvin Jr. et al. in Children’s Cancer Group Protocol 9942 included preirradiation cisplatin-based CTH in patients with residual tumour. The benefit of systemic treatment was observed only in children who had near-total resection (at least 90% of the tumour removed), with event-free survival comparable with those after radial resection and RTH alone (55% vs. 58%) [16]. Venkatramani et al. in the “Head Start” III prospective clinical trial investigated the possibility of deferring RTH in young children by the use of CTH combined with autoHSCT. This approach appeared ineffective in the case of patients with infratentorial tumours in the absence of RTH [13]. Similarly, other authors did not find adjuvant CTH to be effective in improving patients’ outcome, with worse outcomes in those who had the delay in RTH delivery [20,33,34,37,39,46]. Merchant et al. also observed that such children had inferior outcomes, and, in the CTH group, the progression of the disease occurred in 60% patients during systemic therapy, and both PFS and OS were far better in the no-CTH group [25]. In his prospective study of 153 patients, similar outcomes were observed with better local control, event-free survival and OS in patients with localized ependymoma in the no-CTH group [7]. The Children’s Oncology Group trial ACNS0121 led by Merchant et al. confirmed that the outcome of children younger than 3 years of age who received immediate postoperative RTH and for older patients is similar. The total dose of 54 Gy is sufficient to achieve local control in the case of patients under 18 months old [6,46]. The results of the prospective German brain tumour trials HIT-SKK 87 and 92 also showed that delaying RTH jeopardizes survival even after intensive CTH, with 3-year OS of 67% for RTH group compared to only 39% for those without RTH [19]. We did not observe the benefit of adding CTH to the treatment and what is more—looking at the sequence of primary therapy—the outcome of patients who started the adjuvant treatment with CTH instead of RTH was inferior.

All of this strongly indicates that the delay in adjuvant RTH should be avoided since is associated with higher local relapse rate and worse survival. The latest reports concerning the outcome of molecularly defined childhood posterior fossa ependymomas showed that omission of upfront radiation was the strongest predictor of poor outcome, followed by the extent of resection, irrespective of molecular tumour subgroup [49]. CTH, if considered necessary as part of the multimodal therapy, could follow the irradiation.

### 4.4. Age of the Patients

Some studies found that young age could be a poor prognostic factor, as a result of a delay in diagnosis due to non-specific signs/symptoms, more aggressive tumour biology, and, what is the most important—the delay or avoidance of adjuvant RTH due to concerns of possible unacceptable toxicity [11,20,23,29,31,34,37,50]. Although authors who implemented such intensive treatment (surgery combined with immediate RTH) observed very good outcomes [6,7,21,26,31], Snider et al. observed that 10-year OS was increased from 40% without RTH to 66% with RTH and from 43% to 51% in patients with grade III and grade II tumours, respectively [21]. Children in the Merchant et al. prospective study were prescribed a td of 59.4 Gy, except those under the age of 18 months who achieved gross-total resection. The observed 7-years OS of 82% and 80% for children over and under 3 years old, respectively, showed that very good results could be obtained in the youngest children without the avoidance of irradiation in primary setting [7]. Similar results were observed in his next project—the Children’s Oncology Group trial ACNS0121—which proved that the outcome of children with ependymoma younger than 3 years of age who received immediate postoperative RTH and for older patients is similar [6]. In contrast, in the study of Ailon et al., children younger than 2 years old who had R0 resection not followed by RTH had 5-year OS of only 20% and salvage treatment was unsuccessful in all of them [29]. In our group, 5- and 10-year OS and PFS of children less than 3 years old was 86% and 71% for OS, and 54% and 30% for PFS, respectively. RTH was part of the primary treatment in 10 of 17 young patients, and those who received irradiation had statistically significant improvement in the outcome. As suggested in trial ACNS0121, the td of 54 Gy in 1.8 Gy fd should be delivered as part of primary treatment in children between 18 and 36 months old.

### 4.5. Spinal Cord Tumours

Spinal cord ependymomas in children are rare neoplasms. Classical GII and myxopapillary GI ependymomas are the most commonly diagnosed subtypes. Maximal safe resection is the cornerstone of the management of these patients since it improves survival, although it is not always possible due to the possibility of neurologic deficits [50]. The role of irradiation is debatable, and no prospective trials have evaluated its role, either in paediatric or in the adult population, although many authors advocate its use, especially in cases of non-radical resection [24,51,52]. The previously used craniospinal irradiation is no longer a standard treatment unless dissemination along the spinal axis is diagnosed. A very good outcome after R0 resection alone was reported in the study of Yang et al. [50]. Similarly, in a retrospective SEER analysis of 64 children with grade II spinal ependymoma, a very good outcome was achieved with R0 resection alone, with a 10-year OS of 94%. However, patients who received adjuvant RTH after non-radical surgery had better outcome—10-year OS of 87% compared to 75% for the non-RTH group, although the difference was not statistically significant [28]. Data on progression was not available in the SEER database, and some authors reported recurrence in up to 50% of children with myxopapillary tumours, irrespective of the extent of the resection, but only a small rate of recurrences in case of GII tumours after R0 resection (combined in some cases with RTH) [35]. Adjuvant RTH lowered the rate of recurrence in myxopapillary tumours in the study of Pędziwiatr K. et al., with only 2 relapses among 13 irradiated children [32]. Such results were also reported by Zou et al., who found that there might be some evidence for a beneficial effect of RTH after non-radical surgery in paediatric grade II/III intraspinal tumours [51]. However, some authors observed a positive influence of RTH in patients after radical resection with no impact of the relapse risk after subtotal surgery [24]. Despite R0 resection, other factors found to be associated with better outcome are older age and non-anaplastic histology, similarly as in cases of cranial tumours [50]. The results observed in our group could suggest that patients with spinal tumours could benefit from local irradiation, although two out of three relapses occurred in the RTH group after local irradiation in patients with grade II tumours after non-radical resection. Recurrences were observed despite high td used (49.5 Gy and 56 Gy) in the adjuvant setting. The role of RTH in case of children with spinal ependymoma is still debatable, with possible positive influence of local irradiation in patients with myxopapillary tumours, after non-radical resection in grade II patients and in all patients with grade III tumours. The td delivered to the tumour/tumour bed should be in the range of 45 to 54 Gy, depending on the location of the tumour. The use of CT/MR imaging fusion in the treatment planning process is strongly suggested [24,32]. Local irradiation in M0 patients is recommended, with craniospinal RTH reserved for patients with disseminated disease.

### 4.6. Toxicity

The idea of delaying RTH to prevent possible neurologic damage was born in the eighties, when old irradiation techniques were commonly used and the majority of patients received craniospinal irradiation [14]. Vaidya et al. in his study observed that the most common side effects of RTH were nausea/vomiting (57%), tiredness/anorexia (26%) and endocrine deficits (20%). Craniospinal irradiation delivered prior to the year 1992 was the most likely to cause long-term complications, reported in six of 16 (38%) patients, which indicates that the probability of toxicity is now lower, since local treatment is now the standard of care [38]. The SFCE study reported the outcome of the combined treatment of 202 children with toxicity evaluation. All the cases of neurologic side effects (18% of the group) were observed after surgery. The most common grade 3 or 4 late effects were cognitive (14%), auditory (5%) and endocrine (2%), with only one patient with a secondary tumour [34]. Similar long-term side effects were reported by other authors, with the most common being vision impairment (45%), psychomotor deficits (25%) and auditory (20%) or endocrine deficits (6%). The high number of visual complications, which were mostly strabismus, authors regarded as a postsurgical rather than a postirradiation effect [11]. Others also observed a high incidence of neurological deficit and/or haemorrhagic or infectious episodes after the surgery, reported in 63 out of 160 patients [10]. The latest study on patients who received RTH with the use of novel techniques and on the smaller field, covering only the tumour bed with margins, reported no decline in neurocognitive outcome evaluated in several psychological tests. The memory test was the only one in which younger children had worse outcome after the irradiation [49]. Proton therapy, with more conformal dose delivery, carries the possibility of lower toxicity, which reflects the outcome reported in the study by MacDonald et al. Only a few patients were diagnosed with posttreatment endocrinopathies, and none of them experienced a major change in height, and only two children experienced hearing loss (both of them received higher doses to their cochlea than the average median dose because of the tumour extension into the foramen of Luschka). The evaluation of neurocognitive deficits showed no statistical difference between pre- and post-treatment levels, which is very important since almost half of the patients were less than 3 years old [40]. One of the limitations of our study is the lack of the evaluation of acute and late toxicity, although due to the large number of missing data we did not decide to include this information in our analysis. However, as reports showed, such good outcome with low toxicity rate could be provided with newer irradiation techniques, so the delay in RTH delivery often associated with jeopardizing the outcome is no longer justified.

It is worthwhile to mention that the use of CTH is also related to neurologic grade 3 or 4 side effects, which were reported in almost 10% of the patients in the study of Cohen et al. Moreover, grade 3 or 4 hearing damage and pulmonary complications was observed in 13% and 9% of his group, respectively. Furthermore, the number of patients who experienced haematologic toxicity (100%), gastrointestinal complications (26%) and infections (55%) was very high [18].

So far, reports of patients who received the local irradiation of spinal tumours showed no growth impairment or other late effects and the observed complications occurred only in those who received RTH on the whole craniospinal axis. Pędziwiatr K. reported that one case of secondary tumour (gliosarcoma) on the border of the irradiation field was observed after the total dose of 45 Gy ten years after the RTH [32]. We did not collect data on the toxicity, but none of the patients with spinal tumour were diagnosed with secondary neoplasm.

In RTH groups, second malignancies occurred in 3% of the patients in the Merchant et al. study and also in those who received systemic treatment without further irradiation [6]. Patteson et al. reported one case of radiation-induced glioblastoma nine years after proton RTH [37]. The number of secondary malignancies reported was higher if repeated RTH was conducted, with 9% found in Tsang et al.’s series of 101 reirradiation [74]. The 5% rate of secondary meningioma observed in our group is slightly higher than in those studies, although this probably reflects the long observation period of some of our patients, who were treated over 30-40 years ago with old RTH techniques.

### 4.7. Study Limitations

The limitations of our study are its retrospective nature, long period of time during which patient were treated and the different treatment modalities applied. Additionally, no unified imaging modality was used for the evaluation of surgical outcome, and some patients in the early years of the study did not have MR. The relatively low number of patients included into the study could be regarded as one of the limitations, but since the data was collected for a very long period of time (1971 to 2019), it reflects the true number of diagnosed and treated paediatric patients with ependymoma in Poland, and furthermore, the group is comparable with the groups described by other authors (see Table A1). The multicentre nature of the study could be regarded as one of the limitations of our study, because of the increased heterogeneity of clinical practice (total dose, target delineation and margins) and the choice of RTH technique. The different schema of the adjuvant RTH and CTH depending on the institutional protocol at the time of the treatment could be regarded as one of the drawbacks, but it gives us the opportunity to evaluate the impact of the treatment sequence with regard to adjuvant RTH and systemic therapy. Furthermore, we did not perform the review of the pathologic diagnosis for the purpose of this study, although a majority of patients included in recent years had tissue samples revised by an independent pathologist who specialized in paediatric neurooncology. This does not apply to those treated in the early years of the study. We are aware that, as reported in the study of Ramaswamy et al., the most impactful biomarker for posterior fossa ependymoma was molecular subgroup affiliation, independent of other demographic or treatment factors. However, we did not analyse the molecular data, because these evaluations were not routinely performed in the Polish population, and we lacked the data for a majority of our patients [63]. The lack of treatment complication evaluation is also one of the limitations of our study; however, data concerning secondary neoplasms was collected. Despite all of the above, this is one of the biggest reports on Polish children with cranial and spinal ependymoma that reflects clinical practise in light of prospective studies.

### 4.8. Future Studies

Future studies should aim to increase local tumour control in ependymoma by increasing the rate of R0 resection, using second surgery when needed and avoiding adjuvant treatment delays. Since the cumulative incidence of local failure remains high, consideration should be given to higher total doses of RTH and in combination with novel systemic agents. The evolution of RTH for childhood ependymoma will continue as investigators search for ways to lower the number of patients who experience local and neuraxis treatment failure. SRS, IMRT, VMAT, proton and particle therapy are methods for further reductions in the dose delivered to nearby healthy tissues without compromising the possibility of delivering high doses on the tumour bed. Aggressive salvage local treatments for recurrent patients can result in improvement in survival.

The introduction of molecular grading into the diagnosis of ependymomas, combined with clinical and histological features, could be used for developing a risk-stratification model for patients and serves as a model for future treatment decisions [33,61,62,63,64,65,66,67]. The identification of those who could benefit from the treatment deintensification or, on the contrary, a more aggressive approach, should be the aim of future prospective trials.

The use of artificial intelligence and deep learning is a matter of discussion in recent years. It could help not only in the detection process and the classification into benign and malignant types, but also during the follow-up of the patient, and could be of value in terms of RTH and tumour and healthy tissue delineation [80,81,82,83,84,85]. Very high accuracy was reported for many of those systems that could improve the diagnostic process in the future and that have the potential to find tumour regions with the most aggressive tissues [85]. Future studies, which would incorporate such a system into clinical practise, are warranted.

## 5. Conclusions

The outcome of children with ependymoma observed in clinical practise confirmed the role of adjuvant irradiation as a factor improving PFS. The sequence of treatment could impact the survival, and the best results are observed in patients in whom RTH immediately follows surgery. Children under the age of 3 have improved survival when RTH is part of primary treatment. The role of CTH remains debatable.

## Figures and Tables

**Figure 1 diagnostics-11-02360-f001:**
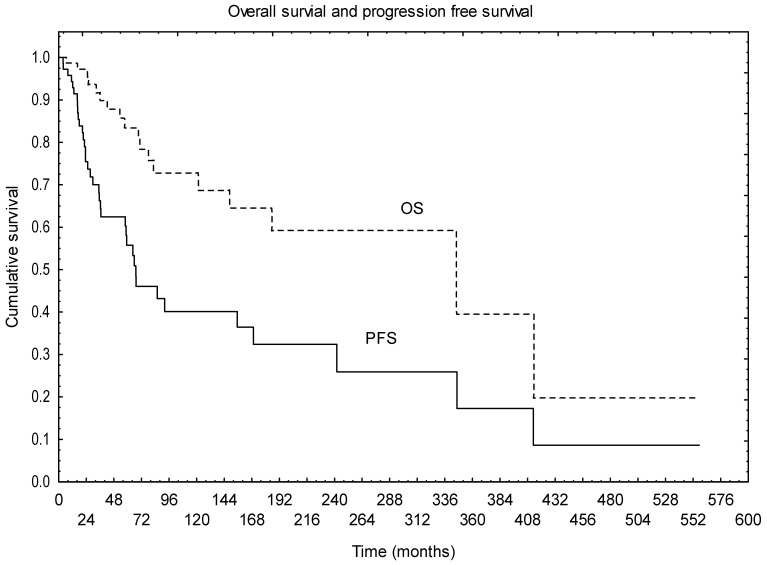
Overall survival and progression free survival of the whole group.

**Figure 2 diagnostics-11-02360-f002:**
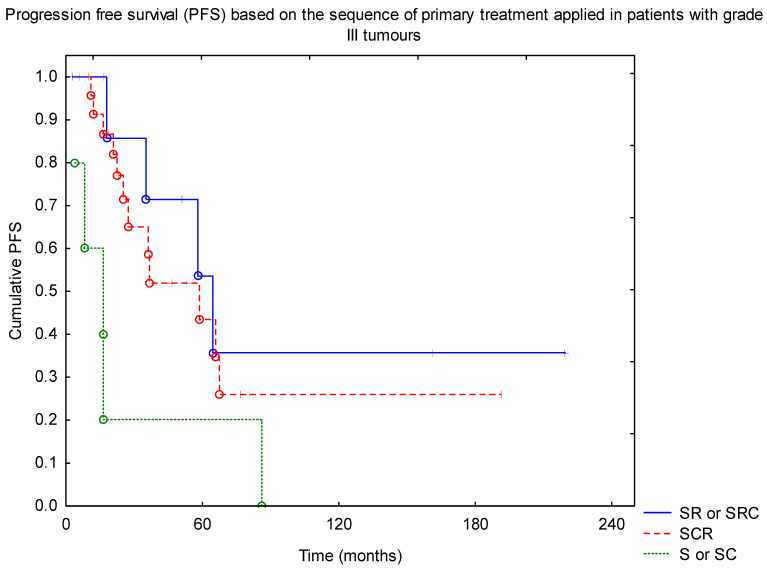
Progression-free survival (PFS) based on the sequence of primary treatment applied in patients with grade III tumours (SRC—surgery + RTH + CTH; SR—surgery + RTH; SCR—surgery + CTH + RTH; S—surgery; SC—surgery + CTH). Abbreviations: CTH—chemotherapy; RTH—radiotherapy.

**Figure 3 diagnostics-11-02360-f003:**
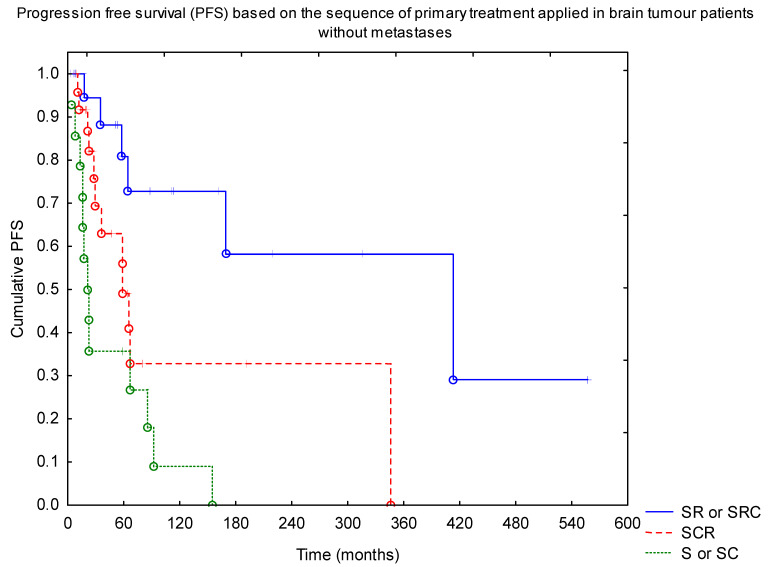
Progression-free survival (PFS) based on the sequence of primary treatment applied in brain tumour patients without metastases (SRC—surgery + RTH + CTH; SR—surgery + RTH; SCR—surgery + CTH + RTH; S—surgery; SC—surgery + CTH). Abbreviations: CTH—chemotherapy; RTH—radiotherapy.

**Figure 4 diagnostics-11-02360-f004:**
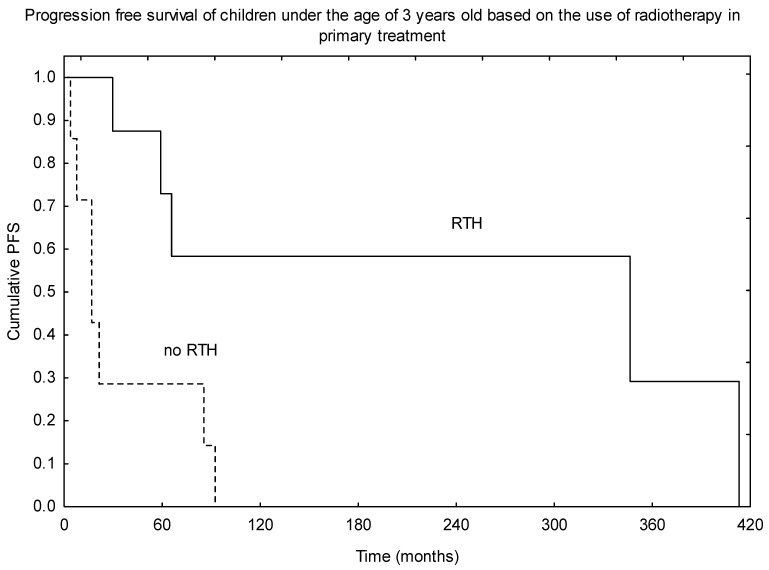
Progression-free survival (PFS) based on the use of radiotherapy (RTH) in the primary treatment in children under the age of 3 years old.

**Table 1 diagnostics-11-02360-t001:** The patients’ and treatment characteristics.

Characteristic		
		Value
Age at diagnosis	Mean (range)	8 (1–18)
		Number of patients (%)
Sex	Female	31 (42%)
	Male	43 (58%)
Pathological subtype	Myxopapillary ependymoma G I Ependymoma G II Anaplastic ependymoma G III	2 (2%) 36 (49%) 36 (49%)
ECOG performance status	0 1 2 3 No data	8 (11%) 53 (72%) 8 (11%) 4 (5%) 1 (1%)
Primary site	Brain Supratentorial Infratentorial No data	66 (89%) 17 (26%) 47 (71%) 2 (3%)
	Spine	8 (11%)
Dissemination at diagnosis	Yes No	7 (10%) 67 (90%)
	Paresis	17 (23%)
	Headaches	34 (46%)
Symptoms	Vomiting/nausea	36 (49%)
	Visual deficits	14 (19%)
	Balance disorders/dizziness	20 (27%)
	Disturbances of consciousness	9 (12%)
	Other	22 (30%)
Underwent surgery	Yes	74 (100%)
	R0 R1 R2/biopsy No data	20 (27%) 24 (32%) 22 (30%) 8 (11%)
Reoperation	Yes No	19 (26%) 55 (74%)
Underwent radiotherapy in primary treatment	Yes Radical Palliative	58 (78%) 57 (99%) 1 (1%)
	No	16 (22%)
Underwent chemotherapy in primary treatment	Yes No	42 (57%) 32 (43%)

Abbreviations: ECOG—Eastern Cooperative Oncology Group; G—grade; R0—radical resection (macro- and microscopically); R1—macroscopically radical resection (but not microscopically); R2—macroscopically not-radical resection.

**Table 2 diagnostics-11-02360-t002:** The radiotherapy characteristics.

Group	Number of pts	TD (Range) [Gy]	Median TD [Gy]	fd (Range) [Gy]	Median fd [Gy]	Local Irradiation/ Craniospinal/ND
Brain tumours	52	45.0–60.0	54.0	1.5–1.8	1.8	25 */25/2
Spinal tumours	6	43.2–56.0	45.0	1.5–1.84	1.67	5/1/0

Abbreviations: fd—fraction dose, Gy—grey, ND—no data, pts—patients, TD—total dose, * 2 patients received local irradiation on ventricular system and tumor bed.

## Data Availability

The data presented in this study are available in this article.

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
