# Peer review of "Polish Multi-Institutional Study of Children with Ependymoma—Clinical Practice Outcomes in the Light of Prospective Trials"

_diagnostics, 2021, doi:10.3390/diagnostics11122360_

Round 1

Reviewer 1 Report

I believe the manuscript is well written and it can add on great knowledge for the readers. But with this, I believe the authors must add some recent trends talk regarding the topic. The literature review seemed to be very concise. The recent brain tumor detection trends should be discussed and for this these manuscripts can be added, 

  • BrainSeg-Net: Brain Tumor MR Image Segmentation via Enhanced Encoder–Decoder Network
  • Cross-modality deep feature learning for brain tumor segmentation

  • Brain tumor segmentation based on deep learning and an attention mechanism using MRI multi-modalities brain images

  • ERV-Net: An efficient 3D residual neural network for brain tumor segmentation

Further, recently a new trend in AI has started which I believe should be discussed in the manuscript, for which following manuscripts can be added, 

  • Novel architecture with selected feature vector for effective classification of mitotic and non-mitotic cells in breast cancer histology images
  • Evolutions and trends of artificial intelligence (AI): research, output, influence and competition

  • Explainable AI and Reinforcement Learning—A Systematic Review of Current Approaches and Trends

Author Response

Thank you very much for your suggestions. However, our article did not concern the topic of brain tumour detection, we included some of mentioned articles in a paragraph “Future studies” in section Discussion and modified it in accordance with the Reviewer suggestions. Unfortunately, we did not have access to all of the mentioned above articles. However, those which were included into the manuscript and concerning the topic of AI and tumour MRI segmentations are inside Reference list (numbers 86 to 91, marked with red font).

Reviewer 2 Report

We read with extreme interest the article by Napieralska entitled “ Polish multi-institutional study of children with  ependymoma – clinical practice outcomes in the light  of prospective trials” where the authors are applying retrospective multi-institutional analysis of 74 children with ependymoma evaluating prospective clinical outcomes in correlation with  different clinical determinants including adjuvant radiotherapy (RTH) and chemotherapy (CTH),   non-disseminated brain tumours, metasatses, resection, Progression,  Recurrence of the disease. survial and progression free survival, toxicity.

The work has a clinical impact to clinical readers and definitely may help shape ependymoma treatment.

There are a few comments on the study:

It would be important if the authors include what kind of organ was impacted by ependymoma metastasis and how it impacted survival rate: is there a correlation?

The data presented in the abstract should be shown in a schematic figure and if possible can be categorized according to each of the multicenter units: these data that need to be included: (All had surgery as primary treatment, with adjuvant 34 radiotherapy (RTH) and chemotherapy (CTH) applied in 78% and 57%, respectively. Median 35 follow-up was 80 months and 18 patients died. Five- and 10-year overall survival (OS) was 83% 36 and 73%. Progression was observed in 32 patients, with local recurrence in 28 cases). This can be added in the supplementary data.

The study should include or comment on the power analysis and limitation in terms of the cases as there may be a bias towards a single center. Please discuss this in the limitation section.

Minor comments:

- autoHSCT was listed at the beginning without being defined and later it was defined: autologous hematopoietic stem cell transplant (autoHSCT)

-survival is spelled wrong: survial and progression

Author Response

We read with extreme interest the article by Napieralska entitled “ Polish multi-institutional study of children with  ependymoma – clinical practice outcomes in the light  of prospective trials” where the authors are applying retrospective multi-institutional analysis of 74 children with ependymoma evaluating prospective clinical outcomes in correlation with  different clinical determinants including adjuvant radiotherapy (RTH) and chemotherapy (CTH),   non-disseminated brain tumours, metastases, resection, progression,  recurrence of the disease, survival and progression free survival, toxicity. The work has a clinical impact to clinical readers and definitely may help shape ependymoma treatment.

There are a few comments on the study:

  1. It would be important if the authors include what kind of organ was impacted by ependymoma metastasis and how it impacted survival rate: is there a correlation?

Thank you very much for your valuable comments. We have looked more carefully at the location of metastases and divided them on two groups: brain (+/- spinal) or solely spinal metastases. We compared those two groups with the log rank test and we did not find the statistically significant impact of the location of metastases on survival (p=0.12). However, all the patients who had only spinal metastases had 5- and 10 year OS of 100%, compared to 62% and 31% for brain metastases group, respectively. Appropriated text was added to the section ‘3.3 Recurrence of the disease’ on page 5 lines 181-183 and marked with the red font: “The most common location of the metastases was brain in 8 patients (four of them had relapse in ventricular system and two multiple metastases also in spinal canal) and spinal canal in 2 patients.” And text: “The location of metastases (brain +/- spine vs only spine) also did not reached statistical significance, however those who had only spinal metastases had 5- and 10 year OS of 100% compared to 62% and 31% for brain metastases group, respectively.” was added to the section ‘3.4. Survival analysis’ on page 7 lines 210-213 and marked with the red font.

  1. The data presented in the abstract should be shown in a schematic figure and if possible can be categorized according to each of the multicenter units: these data that need to be included: (All had surgery as primary treatment, with adjuvant 34 radiotherapy (RTH) and chemotherapy (CTH) applied in 78% and 57%, respectively. Median 35 follow-up was 80 months and 18 patients died. Five- and 10-year overall survival (OS) was 83% 36 and 73%. Progression was observed in 32 patients, with local recurrence in 28 cases). This can be added in the supplementary data.

As suggested by the Reviewer Figure was prepared and pasted in Supplementary data on page 18. (Figure A1).

3.The study should include or comment on the power analysis and limitation in terms of the cases as there may be a bias towards a single center. Please discuss this in the limitation section.

The collaboration between 6 cancer centres has been established to perform this study, so this is not a single centre study. The text “The multicentre nature of the study could be regarded as one of the limitations of our study since increased the heterogeneity of clinical practice (total dose, target delineation and margins) and the choice of RTH technique.” has been already included in section 4.7.Study limitation.

The number of patients collected during the study period (1971-2019) reflects the number of patients diagnosed and treated in clinical practise and it is comparable with the number of patients included into the studies of other authors (in Table A1 15 of 37 cited studies included more patients than we did, but several reports were SEER database analysis and one was included twice as authors updated their results). However, as suggested by the Reviewer, text “The relatively low number of patients included into the study could be regarded as one of the limitations, but since the data was collected for very long period of time (1971 to 2019) it reflects true number of diagnosed and treated patients with ependymoma in Poland, and what’s more, the group is comparable with the groups descried by other authors (see Table A1).” was added to section 4.7. Study limitation on page 14 lines 523-526 and marked with red font.

  1. Minor comments:

- autoHSCT was listed at the beginning without being defined and later it was defined: autologous hematopoietic stem cell transplant (autoHSCT)

-survival is spelled wrong: survial and progression

Corrections, as suggested by the Reviewer, were made.

Reviewer 3 Report

In the paper: "Polish multi-institutional study of children with ependymoma - clinical practice outcomes in the light of prospective trials" the Authors described a cohort of 74 patients treated for ependymoma across several years and several Institution. This is the main limitation of the study.

I have more than one concern that need major revision, about this paper and I try to summarize by points:

  • Literature: 35/68 papers the Authors cited were published before 2015; no paper they cited are published in 2020 and just one was published in 2021; I think they need to update the literature; moreover the Authors need to check the references according to indication of the journal
  • About population: the Authors don't add any details about the protocols used in the patients; since the grading of ependymoma especially in children is very challenge, a revision of the tumors by an indipendent pathologist is necessary to better assess the type of tumors also according to upcoming classification of brain tumors. This is important since the recent literature shows a different results respect the Authors about the grading and the prognosis
  • No biological data about the tumors are reported. Biology of the ependymoma, is not discussed in the paper
  • The discussion is very long: I suggest to summarize better the results of the literature focusing the attention on: surgery because this point is not properly discuss; radiation: the cited a lot of papers about this points but they do not discusse about the role of boost in residual tumors or the role of re-irradiation in the treatment of relapse; chemotherapy: the Authors do not cite the scheduled demonstrated the better outcome; new perspectives: the Authors do not mention nothing about new perspectives (except about proton beam therapy)
  • Spinal cord tumors deserved a separate evaluation since the outcome is very different. Moreover the Authors need to consider that in the upcoming classification also mixopapillary ependymoma are considered grade 2 tumors

To summarize: in my opinion is valuable the description of a national casistic about this particular tumors but the paper needs improvement to avoid messages that are not lined up with the more recent literature.

Author Response

The Reviewer 3 wrote:

In the paper: "Polish multi-institutional study of children with ependymoma - clinical practice outcomes in the light of prospective trials" the Authors described a cohort of 74 patients treated for ependymoma across several years and several Institution. This is the main limitation of the study.

Thank you for your comment. We were aware that this could be regarded as one of the limitations of our study, so the text “The multicentre nature of the study could be regarded as one of the limitations of our study since increased the heterogeneity of clinical practice (total dose, target delineation and margins) and the choice of RTH technique. The different schema of the adjuvant RTH and CTH depending on the institutional protocol at the time of the treatment could be regarded as one of the drawback, but it give us the opportunity to evaluate the impact of the treatment sequence with regard to adjuvant RTH and systemic therapy.” was already added to primary version of the article. However, such data collections reflects true number of diagnosed and treated paediatric patients with ependymoma in Poland, and what’s more, the group is comparable with the groups descried by other authors. Comment on that, as suggested by the Reviewer, has been added to the section Study limitations on page 14 and marked with red font.

I have more than one concern that need major revision, about this paper and I try to summarize by points:

  • Literature: 35/68 papers the Authors cited were published before 2015; no paper they cited are published in 2020 and just one was published in 2021; I think they need to update the literature; moreover the Authors need to check the references according to indication of the journal

Additional literature review was performed. A list of new articles included into the discussion and then in Reference list is marked with red font. The Reference list was re-checked and corrected according to the indications of the journal.

  • About population: the Authors don't add any details about the protocols used in the patients; since the grading of ependymoma especially in children is very challenge, a revision of the tumors by an independent pathologist is necessary to better assess the type of tumors also according to upcoming classification of brain tumors. This is important since the recent literature shows a different results respect the Authors about the grading and the prognosis

Thank you for your valuable comment.

Firstly, the current national protocol for treatment of children with ependymoma was released in 2003 and did not changed so far. The standard treatment in patients with cranial grade III ependymoma includes: maximal safe resection followed by 4 cycles of chemotherapy (etoposide 150 mg/m2 days 1, 2, 3; 42, 43, 44; ifosfamide 3 g/m2 days 1, 2, 3, 21, 22, 23, 42, 43, 44, 63, 64, 65; adriamycin 20mg/m2 days 21, 22, 23, 63, 64, 65) adjuvant radiotherapy and 8 more cycles of chemotherapy (vincristine - 1,5 mg/m2 days 1, 8, 15, lomustin - 75 mg/m2 day 15; cisplatin - 75 mg/m2 day 15). However, as study period covered years 1971- 2019 patients were treated according to the different protocols and information on many of them was unavailable. What is more, since the role of systemic treatment is debatable, based on the results of recent studies and the importance of immediate irradiation which was found to improve survival, some hospitals omitted chemotherapy and implemented radiotherapy after the surgery. The most commonly used systemic agents were etoposide and ifosfamide. The information about protocol and systemic agents used in patients was included in section 3.2. Primary treatment as suggested by the Reviewer on page 4 and marked with red font.

Secondly, we are aware that correct and comprehensive diagnosis is the one of most important steps in the treatment of patients with brain tumours. The majority of patients treated in our institution had tissue samples revised by the independent pathologist specialized in paediatric neurooncology. What is more, the majority of patients treated in other hospitals also had a confirmation of the diagnosis performed by the second pathologist, usually specialised in brain tumours. However, this procedure was not implemented for all of the patients, especially for those in the early years of the study. As suggested by the Reviewer, appropriate text was added to the sections 3.1. Group characteristics page 3 and 4.7 Study limitations pages 14-15 and marked with red font.

  • No biological data about the tumors are reported. Biology of the ependymoma, is not discussed in the paper

Biological data was not available for the majority of patients, thus we did not decide to include it into the analysis. Unfortunately, molecular data evaluation was not routine practice in Polish population and such information was already included in section 4.7 Study limitations. We are aware, that this is one of the major drawbacks of the study but we were not able to collect the tissue samples and evaluate them on the purpose of this study due to legal, logistic and financial reasons.

The biology of ependymomas was shortly described in the section Discussion at the pages 9-10 and included into the section Future studies page 15, as suggested by the Reviewer. All the changes in manuscript were marked with red.

  • The discussion is very long: I suggest to summarize better the results of the literature focusing the attention on: surgery because this point is not properly discuss; radiation: the cited a lot of papers about this points but they do not discusse about the role of boost in residual tumors or the role of re-irradiation in the treatment of relapse; chemotherapy: the Authors do not cite the scheduled demonstrated the better outcome; new perspectives: the Authors do not mention nothing about new perspectives (except about proton beam therapy)

The discussion was corrected as suggested by the Reviewer. Firstly, paragraph concerning the biology was added. Secondly, the parts concerning the surgery, radiotherapy, reirradiation and systemic treatment were added or modified. We did not observe the benefit of adding chemotherapy to the treatment and, what is more – looking at the sequence of primary therapy – the outcome of patients who started the adjuvant treatment with chemotherapy instead of radiotherapy had inferior outcome, so it is hard to provide the schedule with better outcome. Also, the section 4.8. Future studies was modified in accordance with the Reviewer suggestion. All the changes were marked with red font (pages 9 – 15).

  • Spinal cord tumors deserved a separate evaluation since the outcome is very different. Moreover the Authors need to consider that in the upcoming classification also mixopapillary ependymoma are considered grade 2 tumors

Firstly, we conducted more extensive analysis of the group of patients with spinal tumours. Additional information concerning solely the group of patients with spinal tumours was added to section 3.1. Group characteristic (red font). The description of the treatment applied in case of patients with spinal tumours, their outcome and analysis is on pages 4,5,6 and 9 (marked with red font). The RTH characteristic of patients with spinal lesions is also a part of Table 2. As this classification is not available so far, and, we are not able to perform the evaluation of tissues samples from patients included into this study, we did not added the information regarding future changes in upcoming classification into manuscript text.

To summarize: in my opinion is valuable the description of a national casistic about this particular tumors but the paper needs improvement to avoid messages that are not lined up with the more recent literature.

Round 2

Reviewer 3 Report

Thank you to the Authors for accepting my suggestions.

They responded to all of my concerns